# Laboratory Evaluation on Performance of Eco-Friendly Basalt Fiber and Diatomite Compound Modified Asphalt Mixture

**DOI:** 10.3390/ma11122400

**Published:** 2018-11-28

**Authors:** Yongchun Cheng, Di Yu, Yafeng Gong, Chunfeng Zhu, Jinglin Tao, Wensheng Wang

**Affiliations:** 1College of Transportation, Jilin University, Changchun 130025, China; chengyc@jlu.edu.cn (Y.C.); yudi16@mails.jlu.edu.cn (D.Y.); wangws17@mails.jlu.edu.cn (W.W.); 2College of Transportation, Jilin Jianzhu University, Changchun 130021, China; zcf-mine@163.com; 3Jiangxi Transportation Institute, Nanchang 330200, China; taojinglinok@163.com

**Keywords:** laboratory evaluation, diatomite, basalt fiber, compound modify, asphalt mixture

## Abstract

This study proposed an asphalt mixture modified by basalt fiber and diatomite. Performance of diatomite modified asphalt mixture (DAM), basalt fiber modified asphalt mixture (BFAM), diatomite and basalt fiber compound modified asphalt mixture (DBFAM), and control asphalt mixture (AM) were investigated by experimental methods. The wheel tracking test, low-temperature indirect tensile test, moisture susceptibility test, fatigue test and freeze–thaw cycles test of four kinds of asphalt mixtures were carried out. The results show that the addition of basalt fiber and diatomite can improve the pavement performance. Diatomite has a significant effect on the high temperature stability, moisture susceptibility and resistance to moisture and frost damage under freeze–thaw cycles of asphalt mixture. Basalt fiber has a significant effect on low-temperature cracking resistance of asphalt mixture. Composed modified asphalt mixture has obvious advantages on performance compared to the control asphalt mixture. It will provide a reference for the design of asphalt mixture in seasonal frozen regions.

## 1. Introduction

Asphalt pavement has been widely used in the world, but it is affected by the action of vehicle loading and complex natural environment factors [1,2]. The seasonal frozen regions account for 53.5% total area of China and has complex climatic conditions [3,4,5], including high temperatures in the summer, low temperatures in the winter, freeze–thaw (F–T) cycles in the autumn, winter and spring. Asphalt mixture, as a kind of viscoelastic material, suffers rutting deformation at high temperature and pavement cracking at low temperature. What’s more, asphalt mixture has short service life under F–T cycles and suffers fatigue damage due to the action of vehicle loading. Therefore, to address common distresses of asphalt pavement, researchers have been trying to modify asphalt mixture with additives [6,7,8].

Diatomite is a mineral filler with low cost, relative abundance, large specific surface area, and high absorptive capacity [9,10]. What’s more, as an inorganic modified material, diatomite is more environmentally friendly than polymer modifier. Diatomite has been widely used to modify asphalt binder and mixture. Additives can modify the property of asphalt binder, which finally influence the performance of asphalt mixture [11,12,13]. Cong et al. [14] investigated the effects of diatomite on the chemical properties of modified asphalt by Fourier Transform Infrared Spectroscopy test. The results revealed that no chemical reaction occurred between diatomite and asphalt. Cheng et al. [15] indicated that the diatomite modified asphalt had stronger anti-ageing capacity than pure one. Bao [16] studied the pavement performance of diatomite modified asphalt mixture. The results showed that diatomite-modified asphalt mixture had higher Marshall stability, moisture sensitivity, and high temperature stability than pure one. Yang et al. [17] evaluated the modification mechanisms and performance of diatomite modified asphalt mixture. The results showed that the improvement mechanism of diatomite modified asphalt was physical absorption. The bitumen absorption by diatomite brought better resistance in the permanent deformation and moisture damage for asphalt mixture. While tensile strength ratio (TSR) was used as an index to evaluate the moisture susceptibility, the moisture susceptibility had a certain relationship with the resistance to F–T cycles. The addition of diatomite is a possible solution to improve the mechanical properties and F–T resistance for asphalt mixture in the seasonal frozen regions.

Basalt fiber is an eco-friendly mineral fiber with high strength, and low water absorption. In addition, basalt fiber has stable chemical properties when subjected to acid, alkali, and high temperature [18]. Basalt fiber has been widely used to improve the performance of asphalt binder and mixture. Wang et al. [19] investigated the tensile and fatigue property of basalt fiber modified asphalt binder. The results showed that the tensile strength and fatigue life of basalt fiber modified asphalt binder were significantly improved than pure one. Gu et al. [20] studied the effect of basalt fiber on rheological characteristics of asphalt mastic with dynamic shear rheological tests. The results showed that the addition of basalt fiber improved the elasticity characteristics and shear modulus of the asphalt mastic, which meant better high temperature stability. Zhang et al. [21,22] studied viscoelastic characteristics of basalt fiber modified asphalt mortar with a new three dimensional fiber distribution model. The results indicated that the basalt fiber caused stress redistribution and reduced the stress of the asphalt mortar. Zhao [23] and Gao [24] studied the effect of basalt fiber on asphalt mixture, and the results showed that the resistance to low-temperature crack of the mixture was remarkably improved. However, the compound modification effect of the diatomite and basalt fiber on the performance of asphalt mixture is still unclear. Cheng et al. [25] used basalt fiber and diatomite to modify asphalt mastic and found that the high and low temperature properties were improved. Davar et al. [26] studied the properties of basalt fiber–diatomite asphalt mixtures through fatigue test and indirect tensile (IDT) test. But the other mechanical performance and the durability are still unknown. If the addition of diatomite and basalt fibers can give full play to their respective advantages in different performance of asphalt mixtures, and be suitable for the complex environments, then the basalt fiber–diatomite modified asphalt mixture can be well-used to address the common distresses of asphalt pavement in seasonal frozen regions.

In this study, the effect of basalt fiber and diatomite on performance of asphalt mixture were tested and evaluated. A diatomite modified asphalt mixture (DAM), basalt fiber modified asphalt mixture (BFAM), and diatomite–basalt fiber-modified asphalt mixture (DBFAM) and control asphalt mixture (AM) were prepared. Laboratory experiments were carried out to investigate the properties of modified asphalt mixtures. The tests adopted in this research included the wheel tracking test, indirect tensile test at low-temperature, moisture susceptibility test, fatigue test, and F–T cycles test. The pavement performance of modified asphalt mixtures were discussed. It will provide a reference for the design of asphalt mixture in seasonal frozen regions.

## 2. Materials and Methods

### 2.1. Raw Materials

The asphalt AH-90 in this study was obtained from Panjin petrochemical industry, Panjin in Liaoning province, China, and its basic properties are given in Table 1. Basalts were chosen as coarse and fine aggregates, obtained from Wantong Road Building Materials Co., Ltd., Yitong in Jilin Province, China. Their physical properties are shown in Table 2. Limestone powder was chosen as the filler. The selected gradation of asphalt mixture is shown in Figure 1.

Diatomite was used to modify the asphalt mixture, as shown in Figure 2, and its chemical composition are summarized in Table 3. Its pH value is 7.

Basalt fiber was used to modify the asphalt mixture, as shown in Figure 3, and its basic properties are listed in Table 4.

### 2.2. Specimen Preparation

In this study, AM, DAM, BFAM, and DBFAM were produced at the optimum asphalt content (OAC). The contents of modifiers and OAC were determined according to our previous researches [27,28]. The experimental design of these four kinds of asphalt mixtures are listed in Table 5. Modifiers are generally added into asphalt mixture by two ways. One is adding modifier into asphalt, making it dispersed evenly with high-speed shear method, and preparing modified asphalt mixture with modified asphalt. The other is adding the modifier in the mixing process of asphalt mixture. The second method is more convenient. In this research, two methods were tested to find the appropriate method. The prepared modified asphalt by first method and modified asphalt mixture by second method were observed by electron microscopy (Phenom Scientific Instrument, Shanghai, China), as shown in Figure 4. It can be found that both methods can produce evenly dispersed modified materials. Basalt fiber has been dispersed from bundles to single strains of fiber. So the both methods were found to be reasonable. Considering that the progress is simple and can be easily used in the practical engineering, the second method is finally used to prepare the samples of composite modified asphalt mixture.

The cylinder specimens (diameter of 101.6 ± 0.2 mm, height of 63.5 mm ± 1.3 mm) were prepared in accordance to the compaction method of Chinese national standard (JTG E20-2011) [29], which were compacted in a Marshall Compactor (San Yu Lu Tong Instrument Co., Ltd., Beijing, China) with 75 blows on each side. Three identical samples were prepared for each kind of asphalt mixture in each test. The preparation procedures of pure asphalt mixture were conducted as the standard regulates. In order to disperse the basalt fiber into asphalt mixture, the aggregates and the basalt fiber were mixed at temperature of 160 °C for 60 s, then the asphalt was added and mixed for 90 s. The diatomite was added with mineral filler and mixed for 90 s.

### 2.3. Experimental Procedures

#### 2.3.1. Wheel Tracking Test

The pavement in the seasonal frozen regions suffers a high temperature environment in the summer, so adequate high temperature performance of the mixture is required. The wheel tracking test is a widely adopted method to evaluate the high-temperature stability of asphalt mixture [30]. The square slab specimens (length and width of 300 mm, thickness of 50 mm) made of four kinds of asphalt mixtures were prepared in accordance to the Chinese standard [29], which were compacted by rolling compactor for 24 times. Three identical samples were used for each kind of asphalt mixture. The loading time of solid rubber tire applied on the specimen was 60 min with a pressure of 0.7 MPa and running speed of 42 cycle/min. The temperature of the test was 60 °C. The Dynamic stability (*DS*) was used to analysis the high temperature performance of modified asphalt mixture. What’s more, permanent deformation was also used in this paper to evaluate the total deformation in whole process. The *DS* can be calculated by the following equation:(1)$$DS=\frac{(t_{2}-t_{1})\times N}{d_{2}-d_{1}}\times C_{1}\times C_{2}$$ where *DS* is the dynamic stability, in times/mm; *d*_1_ and *d*_2_ are the deformation at time of *t*_1_ (45min) and *t*_2_ (60 min), in mm; C_1_ is the equipment coefficient, which is 1.0; *C*_2_ is the specimen coefficient, which is 1.0; *N* is the round trip speed of wheel, which is 42 times/min.

#### 2.3.2. The low Temperature Indirect Tensile Test

The pavement in the seasonal frozen regions suffers a low temperature environment in the winter, so the low temperature performance of the mixture is required. The low-temperature indirect tensile test was a widely used method to evaluate the crack resistance of modified asphalt mixture [31,32]. To control the temperature, the specimens were placed at −10 °C for 5 h, and an environmental chamber was used in the test, as shown in Figure 5. The loading rate was 1 mm/min. The load and vertical deformation on the top surface of the specimen were recorded. The IDT strength and the IDT failure strain were used to evaluate the low-temperature performance of modified asphalt mixture, which can be calculated by following equations:(2)$$R_{T}=0.006287P_{T}/h$$
(3)$$\varepsilon_{T}=X_{T}\times (0.0307+0.0936\mu )/(1.35+5\mu )$$
(4)$$X_{T}=Y_{T}\times (0.135+0.5\mu )/(1.794-0.0314\mu )$$ where $R_{T}$ is the IDT strength, in MPa; $\varepsilon_{T}$ is the IDT failure strain; $P_{T}$ is the IDT failure load, in N; $Y_{T}$ is the vertical deformation, in mm; $X_{T}$ is the horizontal deformation, in mm; μ is the Poisson ratio, which is 0.25 at −10 °C; and h is the height of specimen, in mm.

#### 2.3.3. Moisture Susceptibility Test

The immersion Marshall test and the freeze–thaw splitting test were the widely adopted methods to evaluate the moisture susceptibility of asphalt mixture [33]. The specimens in treated group of immersion Marshall test were immersed in water for 48 h at 60 °C. The specimens in controlled group were only immersed in water for 30 min at 60 °C. The Marshall stability of all specimens were tested, and the residual stability (MS_0_) was calculated and used as evaluation index of moisture susceptibility. The freeze–thaw splitting test was similar as immersion Marshall test. The specimens in treated group were vacuumed in water, kept at −18 °C for 16 h and thawed in water for 24 h at 60 °C. After that, all specimens in two groups were placed in water bath for 2 h at 25 °C. The tensile strength ratio (TSR) were calculated as index of moisture susceptibility.

#### 2.3.4. Fatigue Test

Fatigue property of asphalt mixture can reflect the service life of asphalt pavement under vehicle loading. Four-point bending fatigue test was widely used to evaluate the performance of modified asphalt mixture [33,34]. This test was conducted according to EN 12697-24D standard. The NU-14 fatigue testing machine produced by Cooper Research Technology Ltd., Ripley, United Kingdom was used. Slab of 400 × 300 × 75 mm was molded and then compacted under a vibration rolling compactor (Earth produce China Ltd., Guangzhou, China). The slabs were cut into beams of 380 × 63 × 50 mm with the cutting machine (Zhongyu Shunfeng Stone Market Co., Ltd., Changchun, China). Fatigue testing machine and vibration rolling compactor are shown in Figure 6. The samples are shown in Figure 7. Strain control loading mode was adopted to study the fatigue life of the asphalt mixture under a microstrain of 800 μm at 20 °C. In addition, sine wave-shaped was applied at frequency of 10 Hz. The stiffness were recorded in the progress of the test. The total number of loading cycles was regarded as the fatigue life when stiffness reaches the 50% of initial stiffness. The initial stiffness was calculated at the 50th loading cycles after the sine wave reach the 800 μm lever.

#### 2.3.5. Freeze–Thaw Cycles Test

Asphalt pavement suffers severe F–T damage in seasonal freeze regions. The water in the pores of the asphalt mixture will produce frost heave when it freezes and flow into new micro-cracks when ice thaws. The repeated F–T cycles can cause significant damage to the pavement performance [35,36]. The moisture susceptibility test can’t directly reflect the damage caused by F–T cycles. So the F–T cycles test were used in this paper to simulate the environmental condition in seasonal frozen regions. To compare with the moisture susceptibility test, the Marshall stability and Indirect tensile (IDT)strength were also conducted under F–T cycles. The F–T condition was choose as vacuumed in water, freeze for 16h at −18 °C and thaw in water for 8 h at 60 °C. After 3, 6, 9, 12, 15 F–T cycles, the residual stability and tensile strength ratio were calculated, which are similar to moisture susceptibility test. However, the freeze–thaw conditions applied in this test was different from the moisture susceptibility test, in order to reflect the performance of asphalt mixture under the effect of freeze–thaw factor more significantly than moisture factor. So the F–T cycles test was more suitable to simulate the environment of seasonal frozen regions.

## 3. Results and Discussion

### 3.1. Wheel Tracking Test Results

The DS and permanent deformation of four kinds of asphalt mixture are shown in Figure 8 and Figure 9. 

As can be seen from Figure 8 and Figure 9, the addition of diatomite and basalt fiber both improve the *DS* and decrease the permanent deformation. The *DS* of DBFAM is higher than DAM, BFAM and AM, which are increased by 15.2%, 42.4% and 72.7%, respectively, and the permanent deformation of DBFAM is the smallest. The high temperature stability of asphalt mixtures is often affected by many factors. Asphalt is a continuous phase in the mixture and plays an important role in the high temperature performance of the mixture. Diatomite has large specific surface area and high absorptive capacity, which can absorb free asphalt, increase the proportion of structural asphalt and improve the adhesiveness between asphalt and aggregate in the mixture. So diatomite can better improve the high temperature performance of the mixture. The basalt fiber has the effects in improving cohesiveness, toughening, crack arrest, bridging macro-cracks, and asphalt adsorption on asphalt mixture. The dispersed basalt fiber in the asphalt mixture can form a three-dimensional network structure and play a reinforcing role, which improve the high temperature performance of asphalt mixture to a certain extent. What’s more, the void of the Marshall specimens of AM, DAM, BFAM, DBFAM are 4.03%, 4.35%, 3.97%, and 4.29% respectively. The reinforcing role of basalt fiber for forming three-dimensional network structure is much obvious than the disadvantage for the increasing void. Under the combined action of basalt fiber and diatomite, the high temperature deformation resistance of the composite modified asphalt mixture is significantly enhanced. Asphalt mixture is greatly affected by the performance of binder. After adding modified materials, the shear viscosity and glass transition temperature of asphalt may also change, which will affect the high and low temperature performance of asphalt mixture [11,12,13,14]. In the previous study, the performance of asphalt mastic has been tested, and the high temperature performance has been significantly improved. The wheel tracking test results of modified asphalt mixture can mutually corroborated with the results of modified asphalt mastic in previous study [25].

### 3.2. The Low-Temperature Indirect Tensile Test Results

The IDT strength and failure strain at −10 °C of four kinds of asphalt mixture are shown in Figure 10 and Figure 11. The addition of basalt fiber improve in the IDT strength and failure strain. The addition of diatomite improved the IDT strength and decreased the failure strain. The strength and failure strain of DBFAM were higher than that of AM. It is clearly that the basalt fiber can improve the low-temperature performance. But the low temperature performance of DAM seems contradictory. The increase in IDT strength may due to the hardening effects of diatomite on asphalt. Diatomite is a porous material with a large specific surface area. After it is added to the asphalt mixture, the bonding ability of the asphalt mortar is improved due to the hardening effects of adsorption between diatomite and asphalt. As a result, the mechanical properties of the mixture will increase. This result meet well with the study of Yang which shows that the indirect tensile (IDT) strength at 20 °C of diatomite modified asphalt mixture has also increased [6]. However, different evaluation indexes for low temperature performance may obtain different conclusions. The deformability at low temperature is also an important low temperature performance. Due to the hardening effects of adsorption between diatomite and asphalt, diatomite also increase the stiffness modulus of the mixture, and its failure strain is not significantly improved, which is not conducive to the low–temperature crack resistance. Considering the indexes in this paper, it can be concluded that the effect of diatomite on the low temperature performance of asphalt mixture is not obvious.

However, it is important to find a kind of modified material that can effectively improve the low temperature performance of asphalt mixture in seasonal frozen regions of China. For example, Jilin province of China is facing severe low-temperature environment, the lowest temperature can reach −30 °C. Basalt fiber can improve the low temperature performance better because of its toughening and preventing cracks and redistribution of stress in asphalt mixture. Therefore, diatomite has little effect on the improvement of low temperature performance than other modified materials. Basalt fiber can be used to improve the low temperature performance of mixture in seasonal frozen regions. The result of DBFAM reflects that adding basalt fiber can solve the low temperature problem of diatomite-modified asphalt mixture. Under the combined action of basalt fiber and diatomite, the low-temperature performance of the composite modified asphalt mixture is significantly enhanced.

### 3.3. Moisture Susceptibility Test Results

The MS_0_ and TSR for each kinds of asphalt mixture are shown in Figure 12 and Figure 13. The addition of diatomite improve the TSR and MS_0_, which means a clear improvement in resistance to the moisture susceptibility. But the addition of basalt fiber decreases the TSR and MS_0_. It seems that adding basalt fiber would decrease the resistance to moisture susceptibility. It is noted that this problem has also appeared in the research of other fibers modified asphalt mixtures, such as glass fiber [37], polypropylene and aramid fibers [38]. The results may due to the reason that the asphalt mixtures become harder to compact after adding hard fiber, and the void of asphalt mixtures would also increase. Higher voids make BFAM more susceptible to moisture damage than AM. However, DBFAM has a high resistance to rutting and moisture susceptibility than AM. The reinforced adhesion effects of diatomite in asphalt can reduce the moisture damage in the bounding between asphalt mortar and aggregate, and can solve the moisture susceptibility problem of BFAM.

### 3.4. Fatigue Performance Test Results

Fatigue test can reflect the performance degradation under repeated loading and show the effects of basalt fiber and diatomite on fatigue life. The changes of bending stiffness modulus under loading cycles is shown in the Figure 14.

Based on the data in Figure 14, DBFAM has the highest fatigue life of 2582 cycles, followed by DAM of 2300 cycles, BFAM of 2182 cycles, while the AM has the lowest one of 1762 cycles. The fatigue life of DBFAM is 46.5% longer than AM. The fatigue lives of asphalt mixtures in this research are lower than other researches by the high strain levels and shape of the wave [6,26,33]. Because the sine wave-shaped loading was applied in this study. The fatigue damage caused by sine wave-shaped was more serious than that of half-sine wave-shape loading. The bending stiffness modulus decreases under the loading cycles. It can be clearly seen from the figures that the fatigue process of asphalt mixture mainly goes through two stages before the 50% initial stiffness modulus. In the first stage, the bending stiffness modulus of the four materials decrease sharply. In the second stage, the modulus of flexural and tensile stiffness decreased slowly and approach linearity. In the first stage, the sharp decrease of bending stiffness modulus is mainly due to the material restructuring in the test specimens under alternating loads. The second stage is the main stage of fatigue damage of the materials. The fatigue damage of the specimens under alternating loads occurs during the initiation and development of micro-cracks. The diatomite and basalt fiber can both enhance the bounding force and adhesive strength between asphalt and aggregates. So the two kinds of modifier can both improve the fatigue performance of mixture at high strain levels.

### 3.5. Freeze–Thaw Cycles Test Results

The F–T cycles test is similar to the moisture susceptibility test. But this method is more reasonable to reflect the damage caused by the repeated action of water icing and water erosion under the F–T cycle. The changes of stability and IDT strength for different asphalt mixtures after F–T cycles are shown in Figure 15, Figure 16, Figure 17 and Figure 18.

It can be seen from Figure 15, Figure 16, Figure 17 and Figure 18 that the Marshall stability and IDT strength of each type of asphalt mixture is continuously reducing, and the loss ratio of them are continuously increasing under the F–T cycles. The MS and IDT strength of BFAM, DAM, DBFAM are still high than that of AM under F–T cycles. The loss of MS for the DAM and DBFAM are obviously smaller than that of AM. As for the BFAM, the loss is slightly less than AM. The results show that the addition of diatomite can effectively improve the resistance to F–T cycles, while the basalt fiber also has some improvement but not obviously. The results of IDT strength under F–T cycles are the same as that of MS.

The change of performance can show the damage caused by F–T cycles. As the number of F–T cycles increases, the performance of all asphalt mixtures decreases. At the beginning of the F–T cycle, the performance decreases rapidly. After 6 F–T cycles, the performance degradation rate slows down. At the beginning of the F–T cycle, under the effect of freezing, the void of the specimen increases, and the mixture becomes loose. The moisture gradually invades into the internal void and the asphalt film, and the mechanical properties are significantly reduced. As the F–T cycles continues to increase, the voids are gradually connected, which creates communication channel. The pressure generated during the freezing of the void water can be released through this channel. Therefore, the increasing rate of damage is slowed down. The addition of diatomite can reinforce adhesion between asphalt and aggregate, which reduce the moisture damage in the bounding between asphalt mortar and aggregate. It is more difficult for moisture to intrude into internal voids and asphalt film. So diatomite can effectively improve the resistance to F–T cycles. The basalt fiber may also has a little improving due to the bridging and crack arrest effect on limiting the crack propagation during the freezing process. For the effect of basalt fiber on air voids, the freezing effect will increase, but for the toughening and crack resistance effect of basalt fibers on inhibiting the expansion of cracks, and ultimately the resistance to F–T cycles does not decrease. In addition, the diatomite can improve the resistance to F–T cycles. Therefore, the method of composite modification is reasonable. However, The DBFAM has the high resistance to F–T cycles and can be well used in seasonal frozen regions.

## 4. Conclusions

This paper investigated the performance of asphalt mixtures modified by basalt fiber and diatomite. The effect of diatomite and basalt fiber on the performance of asphalt mixture were studied by means of wheel tracking test, low temperature IDT test, moisture susceptibility test, four-point bending fatigue test and F–T cycles test. Based on the above analysis, the following conclusions could be drawn:The addition of basalt fiber and diatomite can both improve the resistance to rutting than AM. DBFAM has highest dynamic stability, which means DBFAM can better resist rutting or permanent deformation at high temperature in asphalt pavements.The addition of basalt fiber can obviously improve the resistance to low-temperature cracking than AM while diatomite is not obviously. DBFAM has high IDT strength and IDT failure strain, which means DBFAM can better resist cracking at low temperature in seasonal frozen regions.The addition of diatomite can increase the resistance to moisture damage while basalt fiber can’t. DBFAM have a higher resistance to moisture susceptibility than AM, which means DBFAM can better resist moisture damage in asphalt pavements.The addition of basalt fiber and diatomite can both improve the fatigue cracking resistance than AM. DBFAM has longest fatigue life than others under high strain lever, which means DBFAM has longer service life under the repeated vehicle loading in asphalt pavements.The addition of diatomite can obviously increase the resistance to F–T cycles while basalt fiber is not obviously. DBFAM have a higher resistance to F–T cycles than AM, which means DBFAM can better resist the freeze–thaw cycles damage in seasonal frozen regions.

## Figures and Tables

**Figure 1 materials-11-02400-f001:**
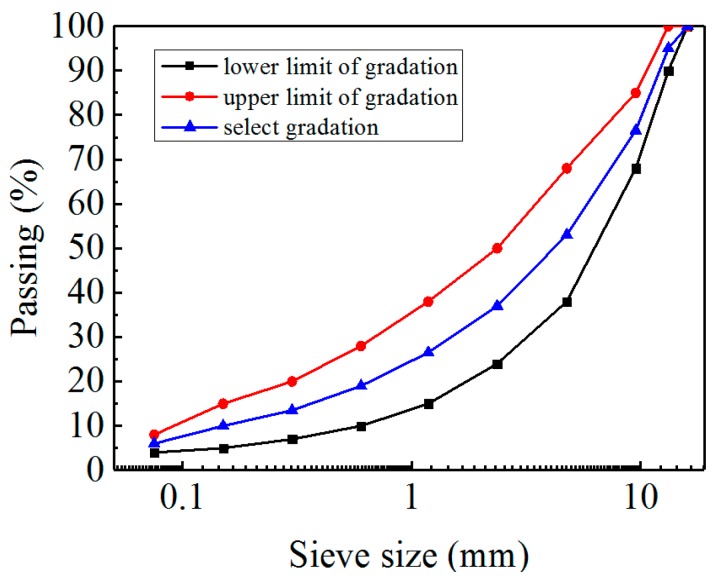
The Grading Curve aggregates used in this study.

**Figure 2 materials-11-02400-f002:**
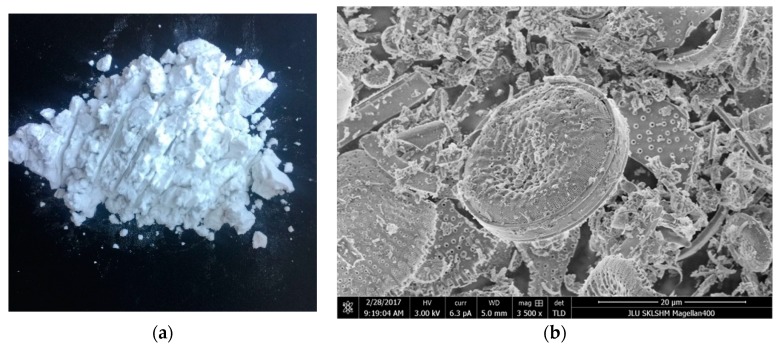
Images of diatomite. (**a**) Morphology of diatomite; (**b**) SEM images of diatomite.

**Figure 3 materials-11-02400-f003:**
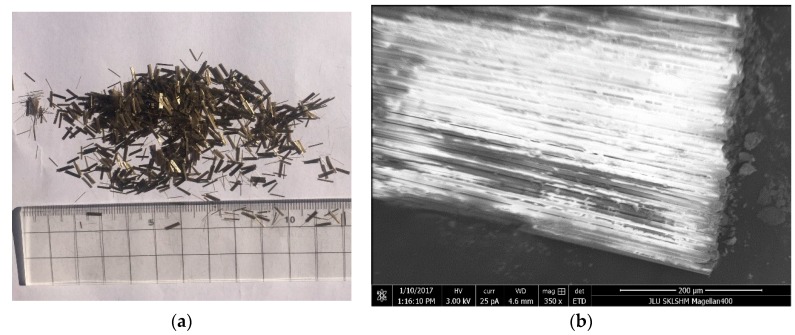
Images of basalt fiber. (**a**) Morphology of basalt fiber; (**b**) SEM images of basalt fiber.

**Figure 4 materials-11-02400-f004:**
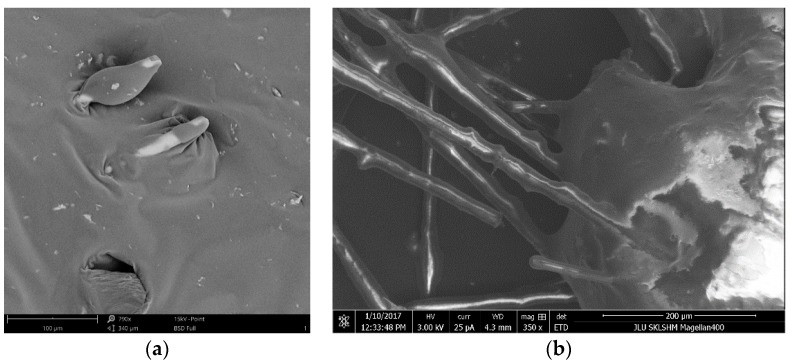
Images of modified asphalt and asphalt mixture. (**a**) SEM images of modified asphalt; (**b**) SEM images of modified asphalt mixture.

**Figure 5 materials-11-02400-f005:**
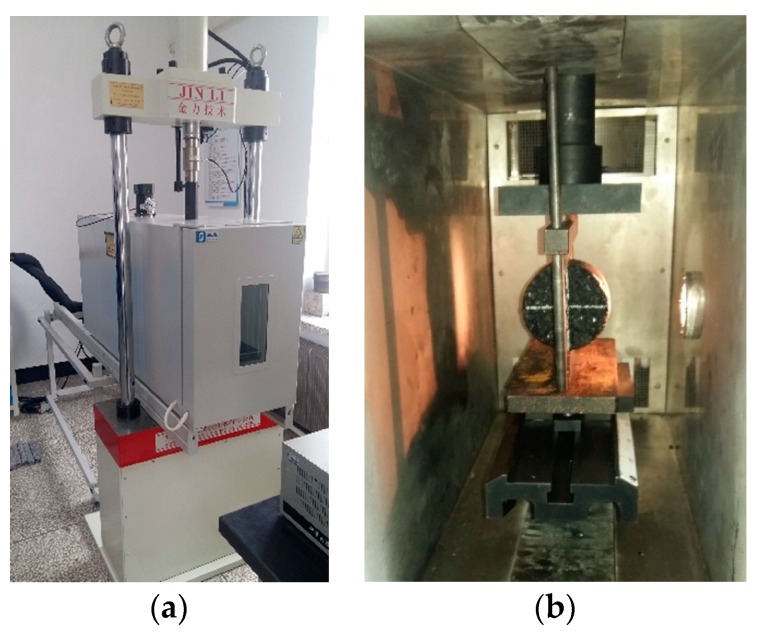
The low temperature indirect tensile test. (**a**) Testing equipment. (**b**) Testing progress.

**Figure 6 materials-11-02400-f006:**
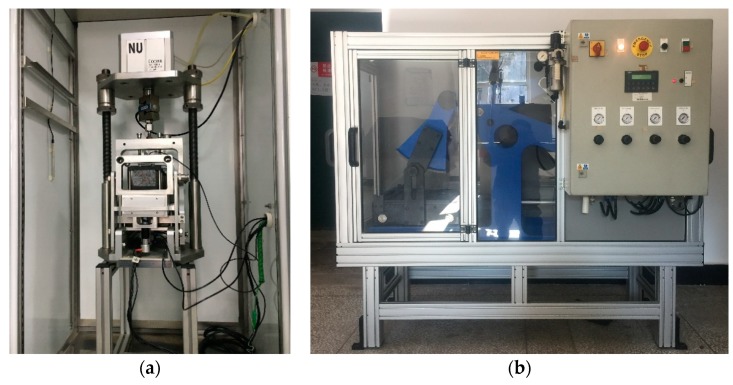
Equipment of Fatigue test. (**a)** Fatigue testing machine; (**b**) Vibration rolling compactor.

**Figure 7 materials-11-02400-f007:**
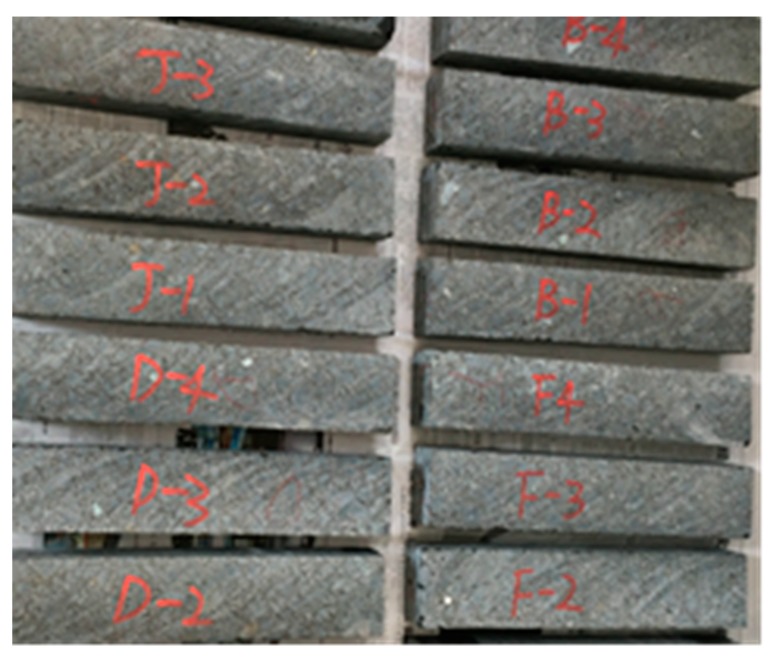
Samples of Four-point Bending Fatigue test.

**Figure 8 materials-11-02400-f008:**
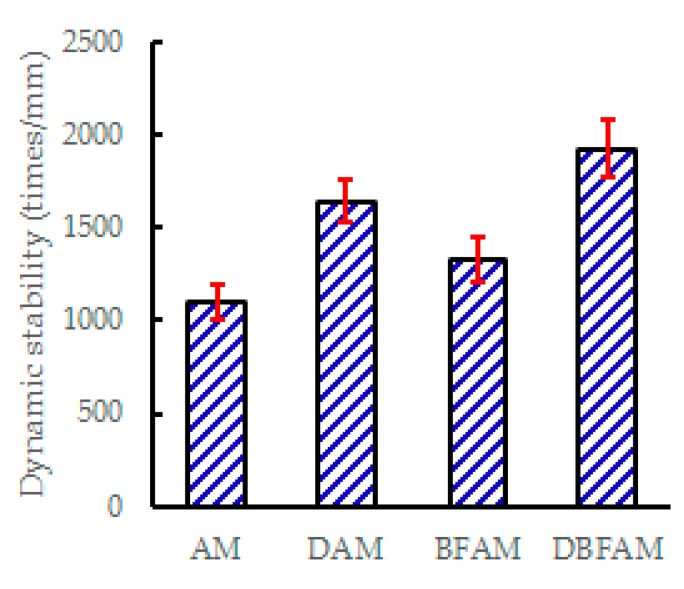
Dynamic stability of asphalt mixtures.

**Figure 9 materials-11-02400-f009:**
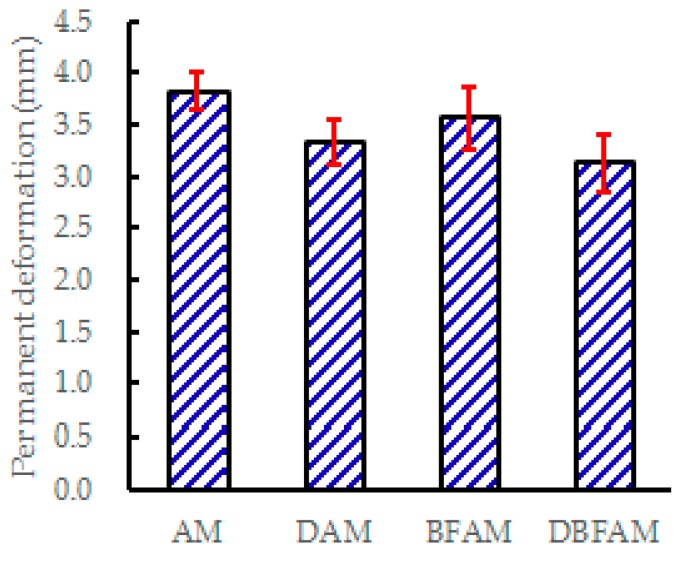
Permanent deformation of asphalt mixtures.

**Figure 10 materials-11-02400-f010:**
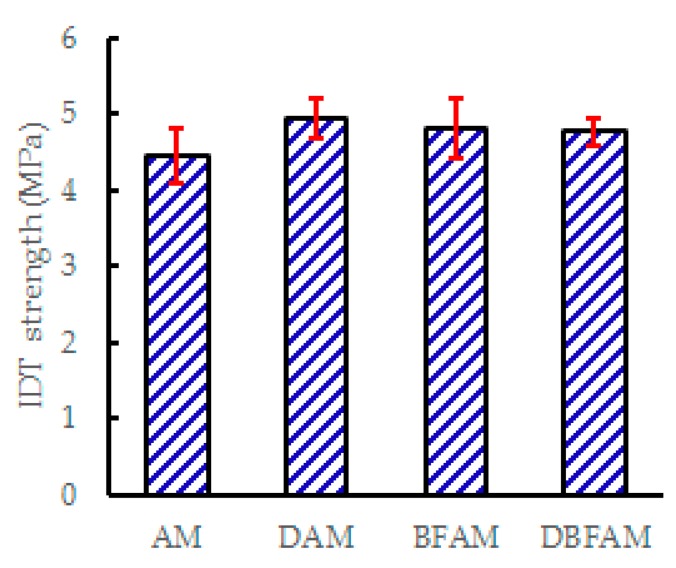
Indirect tensile strength of asphalt mixtures at −10 °C.

**Figure 11 materials-11-02400-f011:**
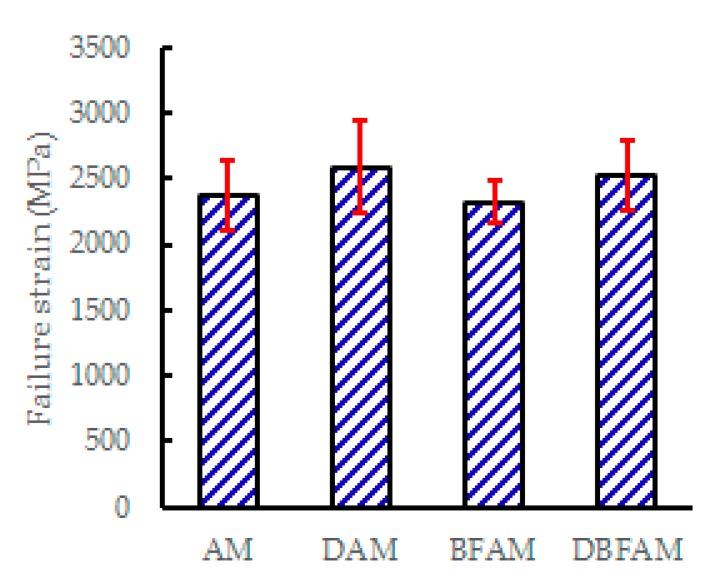
IDT failure strain of asphalt mixtures.

**Figure 12 materials-11-02400-f012:**
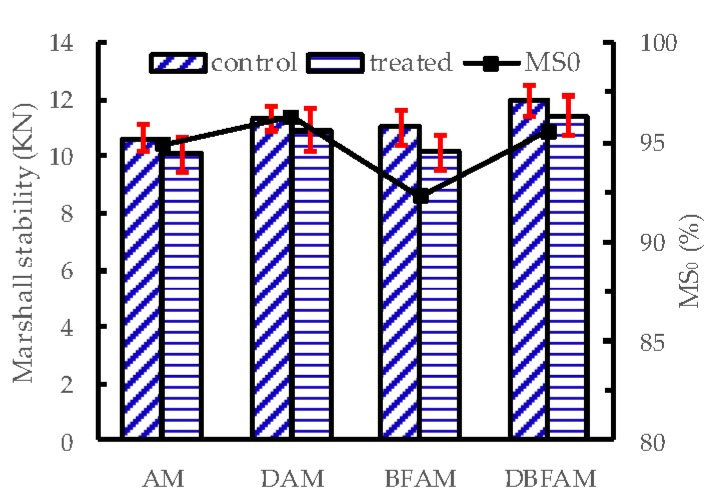
MS_0_ of asphalt mixtures.

**Figure 13 materials-11-02400-f013:**
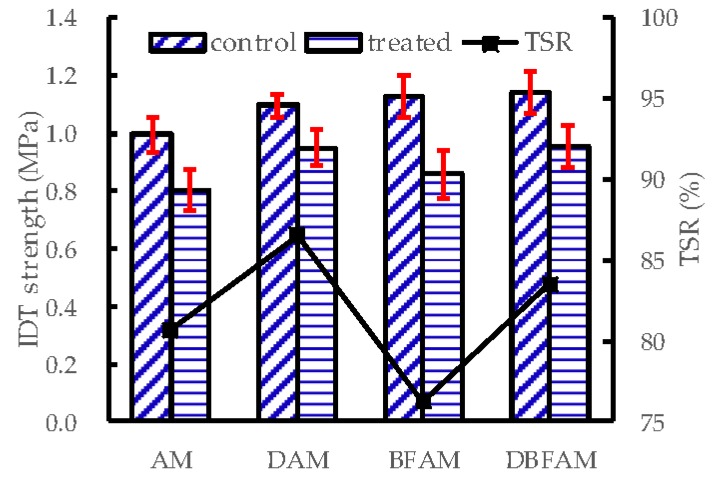
Tensile strength ratio of asphalt mixtures.

**Figure 14 materials-11-02400-f014:**
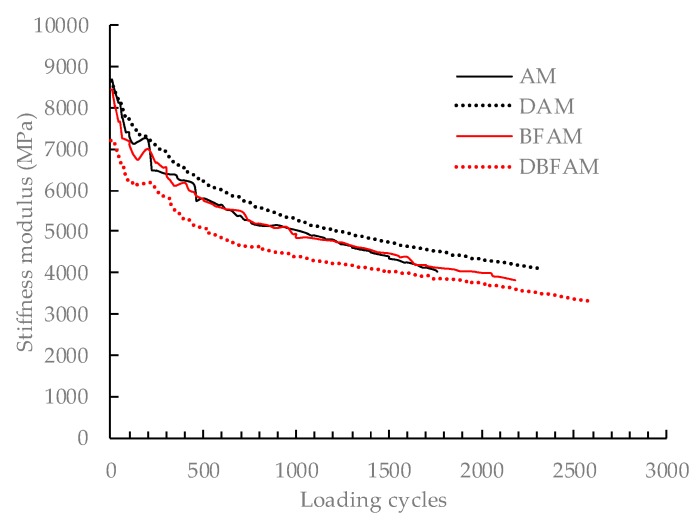
Stiffness modulus under fatigue test.

**Figure 15 materials-11-02400-f015:**
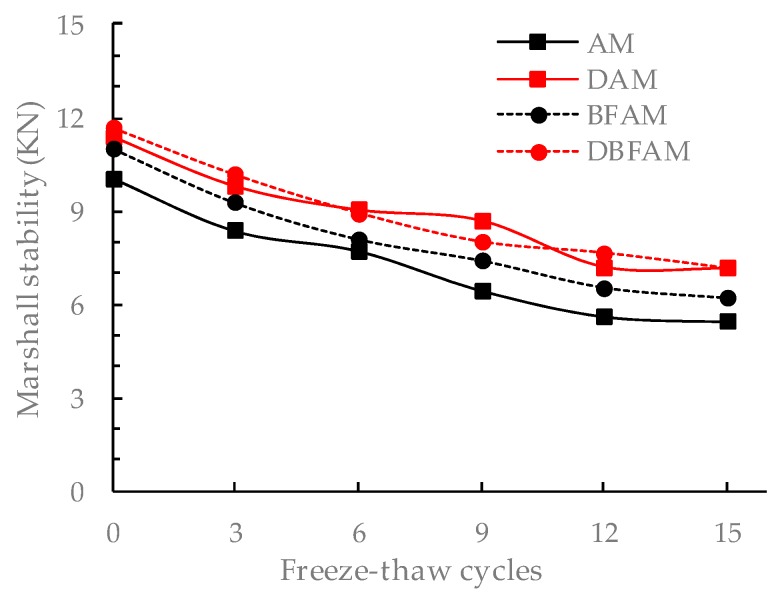
Marshall stability under freeze–thawcycles.

**Figure 16 materials-11-02400-f016:**
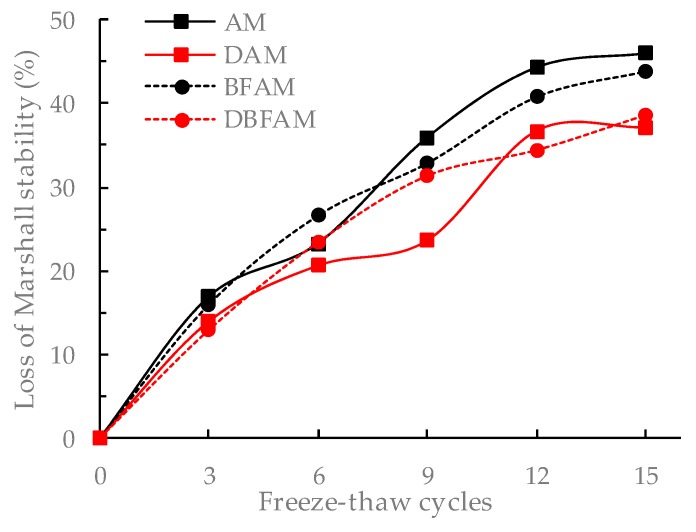
Loss of Marshall stability under F–T cycles.

**Figure 17 materials-11-02400-f017:**
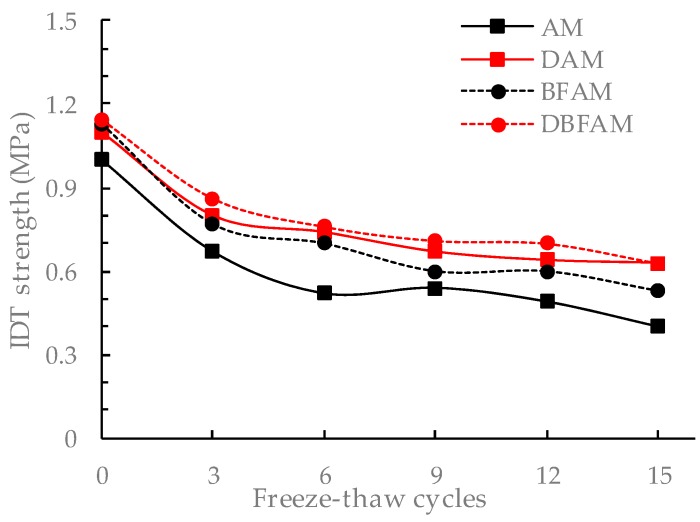
IDT strength under F–T cycles.

**Figure 18 materials-11-02400-f018:**
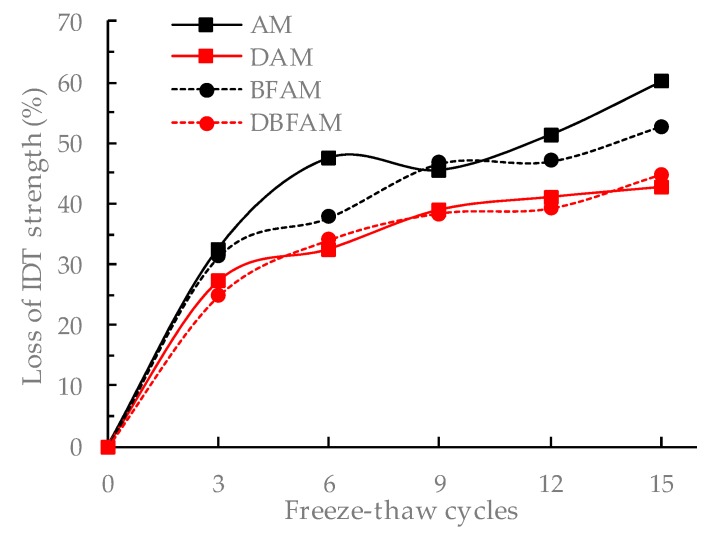
Loss of IDT strength under F–T cycles.

**Table 1 materials-11-02400-t001:** Basic properties of AH-90 asphalt.

Property	Value
Density (15 °C, g/cm^3^)	1.018
Penetration (25 °C, 0.1 mm)	92.3
Softening point (°C)	46.9
Ductility (25 °C, cm)	>150

**Table 2 materials-11-02400-t002:** Properties of aggregate.

Sieve Size (mm)	13.2	9.5	4.75	2.36	1.18	0.6	0.3	0.15	0.075
Crushed stone value (%)	14.9	13.6	16.0	15.8	-	-	-	-	-
Flakiness index (%)	8.8	9.2	9.1	8.3	-	-	-	-	-
Los Angeles abrasion loss (%)	19.0	18.5	19.0	15.7	-	-	-	-	-
Apparent density(g/cm^3^)	2.811	2.805	2.815	2.817	2.808	2.805	2.778	2.777	2.768
Absorption coefficient of water (%)	0.33	0.44	0.54	0.75	-	-	-	-	-

**Table 3 materials-11-02400-t003:** Diatomite chemical composition.

Chemical Composition	SiO_2_	Al_2_O_3_	Fe_2_O_3_	CaO	MgO	TiO_2_	K_2_O	Loss on Ignition
Content (%)	85.60	4.50	1.50	0.52	0.45	0.30	0.67	4.61

**Table 4 materials-11-02400-t004:** Technical properties of basalt fiber.

Property	Test Result	Standard Requirements
Diameter (μm)	10–13	—
Length (mm)	6	—
Moisture content (%)	0.030	≤0.2
Combustible content (%)	0.56	—
Linear density (Tex)	2398	2400 ± 120
Fracture strength (N/Tex)	0.55	≥0.40
Tensile strength (MPa)	2320	≥2000
Tensile modulus of elasticity (GPa)	86.3	≥85
Elongation at break (%)	2.84	≥2.5

**Table 5 materials-11-02400-t005:** Experimental design of asphalt mixtures. AM: control asphalt mixture; DAM: diatomite modified asphalt mixture; BFAM: basalt fiber modified asphalt mixture; DBAFM: diatomite and basalt fiber compound modified asphalt mixture.

Group	Diatomite ^1^ (%)	Basalt Fiber ^2^ (%)	OAC (%)
AM	0	0	4.78
DAM	0	6.5	5.12
BFAM	0.25	0	5.09
DBFAM	0.25	6.5	5.22

^1^ Diatomite content is the replace volume ratio of the diatomite to-entire filler; ^2^ Basalt fiber content is the weight ratio of the basalt fiber to asphalt mixture.

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
