# Peer review of "Laboratory Evaluation on Performance of Eco-Friendly Basalt Fiber and Diatomite Compound Modified Asphalt Mixture"

_materials, 2018, doi:10.3390/ma11122400_

Round 1

Reviewer 1 Report

Detailed comments (mostly language and style) are given in the provided commented PDF file. 

More significant comments are as follows:

-     although abbreviations can be included in the abstract for clarity and ease of text scanning by the potential reader (correct use: "...diatomite modified asphalt mixture (DAM), basalt fiber modified..."  abbreviations should not be referenced in the abstract again i.e.: “…obvious advantages on 20 performances than AM.”

-       When describing results of scientific investigations and describing findings use past tenses;

 -       The use of noun “performance”: when talking about the performance of a material, its characteristics, response to stress etc., “performance” is an uncountable noun, in this context always use singular “performance”.

 -       L31-32: what do the authors mean my “early” F-T cycle damage?-       L46: was It then bitumen absorption or “bonding enhancement” ?-       L134: if all of the samples were compacted by subjecting them to 24 passes of the compactor, were the air voids controlled in the slabs? Was the influence of air void contents in the slabs considered when evaluating the results? DFBAM was surely more difficult to compact than the reference AM;-       L176: there is a number of questions regarding the fatigue testing:

    o    What was the reason to adopt the 800 μm strain level (amplitude)? This magnitude of strain is most probably well beyond linear viscoelastic region of any asphalt mix at 20degC;    

    o    The obtained fatigue results clearly indicate that with beam failure at 2000 – 2600 loading cycles which is far from the limits given by the standard (10 000 – 2 000 000) proving that the adopted strain levels were to high;

    o    According to the standard, the initial stiffness modulus should be calculated after to 100th loading cycle not the 50th;

    o    Were the air void contents controlled in the beam samples?

    o    How do the testing conditions relate to the actual mix performance in the pavement?

-       Fig 8 and 9: what do the error bars represent? The quality of the figures is inadequate and the fill of the bars should be changed for clarity;

-       Fig 10 and 11 and others: what do the error bars represent?

-       p. 3.3: the results are not confronted with air voids which surely and significantly contribute to the observed results;

-       L264: the observed poor fatigue resistance was rather a result of the adopted very high strain amplitude;

-       No statistical analysis is provided therefore significance of the findings is unknown – especially in case of the low temperature performance

Author Response

We thank you very much for giving us an opportunity to revise our manuscript entitled ” Laboratory Evaluation on Performance of Eco–friendly Basalt Fiber and Diatomite Compound Modified Asphalt Mixture” (materials–397397) by Yongchun Cheng, Di Yu, Yafeng Gong, Chunfeng Zhu, Jinglin Tao and Wensheng Wang for publication in Materials.

We are thankful to the Reviewer and Editor for pointing out some important modifications needed in the paper. We have seriously taken into account of these comments. The explanations of what we have changed in response to the reviewer’s comments are given point by point in the following pages attached in this letter.

We hope that all these changes fulfill the requirements to make the manuscript acceptable for publication in Materials.

Looking forward to hearing from you soon.

Sincerely yours,

Yongchun Cheng, Di Yu, Yafeng Gong, Chunfeng Zhu, Jinglin Tao and Wensheng Wang

Reviewer 2 Report

Authors study the effect of fiber addition on the mechanical properties of asphalt and find the high-temperature properties of asphalt are improved upon addition of by basalt fiber and diatomite and combination of these two additives.  In overall, the organization of the paper is acceptable but I have a few comments related to possible rheological tests.  In addition, the text needs to be completely revised in order to eliminate grammar mistakes and typos.  My comments are as follows:

1.       In line 52, authors use “kind of” bio-friendly for basalt fibers.  Please avoid using these vague phrases in the manuscript.

2.       In line 68, authors mention “Arash Davar et. al [14]”, reference [14] is not this one.  Please check the bibliography and citations and make sure they are consistent.  Furthermore, this reference should be cited as Davar et al.

3.       The writing needs to be checked to avoid any possible grammar mistakes: “The tests adopted in this research includes the wheel tracking test, indirect tensile test at low–temperature, moisture susceptibility test, fatigue  performance test and F–T cycles test.
or “The addition of diatomite can increase the resistance to moisture damage while basalt fiber

can’t.

4.       How do these additives change the shear viscosity (or other rheological tests such as LAOS) of asphalt at high temperatures?  Given these materials have an impact on the rutting, I expect the high-temperature rheology should be different for these asphalt samples. 

5.       Authors need to determine the glass transition temperature of their samples.  The effect of additives on the fragility of the system will be visible in the glass transition tests. 

6.       Another important question that authors can address when they perform the rheological tests is that whether the time-temperature superposition principle is valid or not? Some relevant papers that might be of interest for the mechanical and  rheological properties, and glass transition:

** Fallah, F.; Khabaz, F.; Kim Y-R, Kommidi; S. R. and Haghshenas, H; Fuel, 237, 71-80, 2018.
** Anderson, D. A., D. W. Christiansen, H. U. Bahia, R. Dongre, M. G., Sharma, C. E. Antle, and J. , Button, Binder characterization and evaluation , Volume 3: Physical characterization, Technical, Report No., SHRP-A-369, Federal Highway Administration, Washington, DC,1994

Author Response

(The authors gave the same response as above.)
